# Induction of fatigue-like behavior by pelvic irradiation of male mice alters cognitive behaviors and BDNF expression

**Brian S. Wolff, Sumiyya A. Raheem, Sarah A. Alshawi, Jeniece M. Regan, Li Rebekah Feng, Leorey N. Saligan** *

National Institute of Nursing Research, National Institutes of Health, Bethesda, Maryland, United States of America

* saliganl@mail.nih.gov

## Abstract

Fatigue and cognitive deficits are often co-occurring symptoms reported by patients after radiation therapy for prostate cancer. In this study, we induced fatigue-like behavior in mice using targeted pelvic irradiation to mimic the clinical treatment regimen and assess cognitive behavioral changes. We observed that pelvic irradiation produced a robust fatigue phenotype, a reduced rate of spontaneous alternation in a Y-maze test, and no behavioral change in an open field test. We found that reversal learning for fatigued mice was slower with respect to time, but not with respect to effort put into the test, suggesting that fatigue may impact the ability or motivation to work at a cognitive task without impairing cognitive capabilities. In addition, we found that mice undergoing pelvic irradiation show lower whole-brain levels of mature BDNF, and that whole-brain proBDNF levels also correlate with spontaneous alternation in a Y-maze test. These results suggest that changes in BDNF levels could be both a cause and an effect of fatigue-related changes in behavior.

## 1. Introduction

Fatigue is a common symptom associated with cancer and cancer treatments, including chemotherapy and radiation therapy, and it can dramatically impact quality of life [1]. Cancer-related fatigue (CRF) often co-occurs with other symptoms, including cognitive deficits, even when the central nervous system is not directly affected by the cancer or the treatment [2, 3]. It is therefore likely that systemic changes, for example changes in immune response [4], are causing the co-occurrence of symptoms, but the process remains poorly understood. Mechanistic studies of fatigue and its co-occurring symptoms in clinical settings can be challenging, so animal models may be critical to uncover the mechanisms behind CRF and its related symptoms.

A recent study in men receiving radiation treatment for prostate cancer found that the cognitive deficits accompanying CRF were related specifically to response times and not accuracy in a cognitive task [5]. This result resembles findings in patients with chronic fatigue syndrome [6, 7] (CFS), which suggests that this specific cognitive deficit could be a common feature of

**Data Availability Statement:** All data and code are available in an OSF repository: https://osf.io/9237v/ DOI 10.17605/OSF.IO/9237V.

**Funding:** This study was supported by the Divisions of Intramural Research of the National Institute of Nursing Research and the National Institute of Mental Health of the NIH, Bethesda, Maryland.

**Competing interests:** The authors have declared that no competing interests exist.

**Abbreviations:** CRF, cancer related fatigue; CFS, chronic fatigue syndrome; VWRA, voluntary wheel running activity; mBDNF, mature BDNF.

fatigue across clinical populations. To our knowledge, it is unknown whether this generalizes to animal models of CRF.

Brain-derived neurotrophic factor (BDNF) is expressed throughout the brain and involved in various aspects of brain function. Mature BDNF (mBDNF) and its precursor, proBDNF, have opposing actions [8, 9]: mBDNF interacts with tropomyosin receptor kinase B (TrkB) and promotes cell survival, neurite outgrowth, and synaptic long-term potentiation; while proBDNF binds to pan-neurotrophin receptor 75 (p75$^{NTR}$) and promotes apoptosis, growth cone retraction, and synaptic long-term depression. Patients with CRF often exercise less than healthy controls [10], and exercise has been strongly associated with elevated levels of mBDNF in both clinical [4] and animal models [11, 12]. Interestingly, some beneficial effects of exercise appear to be mediated by an upregulation of BDNF in the brain, as well as an enhancement of mBDNF and a reduction of proBDNF [13]. In addition, CRF and related symptoms in human subjects have been associated not only with lower serum/plasma levels of BDNF [14, 15], but also a single-nucleotide polymorphism of the BDNF gene [16, 17]. Based on the literature, fatigue and BDNF may form a feedback loop: fatigue and its consequential decrease in voluntary activity may alter BDNF levels, and BDNF levels may also affect fatigue symptoms. The relationship between BDNF and CRF has not been well-explored, and further study is warranted to understand the connection.

The goal of the current study was to test whether we can observe a relationship between fatigue-like behavior, cognitive behavior, and BDNF expression levels in a pelvic-irradiated mouse model. It is a common question about CRF whether symptoms are driven primarily by the cancer, the treatments, or a combination of the two. Importantly, in this study we used healthy and wild-type male mice, so the effects we describe are specifically attributable to irradiation and not to any interactions with cancer. We used fractionated irradiation targeted to a pelvic region of male mice, which induces a fatigue-like behavior that is defined as a decrease in voluntary locomotor activity.

There are two main parts of the study. First, we tested whether the fatigue-like behavior is accompanied by changes in cognitive or anxiety-like behavior, and we saw significant differences in the rates of both spontaneous alternation and reversal learning when comparing irradiated and sham-irradiated mice. Second, we tested whether whole brain levels of mBDNF and proBDNF are altered by the irradiation procedure, and we found that irradiation had an effect of mBDNF levels, but that spontaneous alternation behavior correlated only with proBDNF levels.

## 2. Materials and methods

### 2.1. Ethics

This study was approved by the National Heart Lung and Blood Institute (NHLBI) Animal Care and Use Committee of the National Institutes of Health (NIH), Bethesda, Maryland, USA. All aspects of animal testing, housing, and environmental conditions used in this study were in compliance with *The Guide for the Care and Use of Laboratory Animals [18]*.

### 2.2. Animals

100 six-week old male C57BL/6NCrl mice were ordered from Charles River Laboratories (Frederick MD) and were individually housed on a 12:12 hour light-dark cycle at roughly 22.2˚C. All mouse handling and experimental procedures were conducted during the light cycle, and, unless otherwise specified, mice had *ad libitum* access to food and water. Three mice died during the study (two failed to wake up from anesthesia, the other for unknown reasons) and their data were removed from all analysis.

## 2.3. Irradiation

This method is described in detail in an earlier publication [19], including the design of the shielding used to target irradiation to the pelvis. In brief, mice were assigned to irradiated ("Irrad") or sham ("Sham") groups so that body weights were evenly distributed between groups; grouping was otherwise random. Once per day for three days, mice were anesthetized with a mixture of 100 mg/kg ketamine (MWI Animal Health, Boise, ID, USA) and 10 mg/kg xylazine (Akorn Animal Health, Lake Forest, IL, USA) and placed inside a lead shielding device within a GammaCell 40 Exactor irradiator (Best Theratronics, Ottowa Ontario, Canada), where they then received 8 Gy irradiation targeted to a pelvic region. This dose causes no overt changes in physical appearance nor signs of tissue damage in the mice [20], though it does induce a decrease in bodyweight [21]. Mice in the Sham group underwent the same procedure as those in the Irrad group, except that they were left outside of the irradiator.

## 2.4. Voluntary Wheel Running Activity (VWRA)

76 mice were housed in cages with running wheels (Lafayette Neuroscience, Indiana, USA) that recorded wheel rotation in one-minute intervals. Mice were transferred into running wheel cages after a week of acclimation to the animal facility. After five days of baseline VWRA recording (days -5 through -1), mice were housed in cages without running wheels for the three days of irradiation (days 0–2). The day after finishing irradiation, animals were transferred back into clean running wheel cages and VWRA was recorded for three more days (days 3–5). VWRA was quantified as the number of minutes during which the wheel rotated. We excluded data from wheels that did not accurately count revolutions or from animals that did not run a consistent amount during the baseline period, which we defined as a 50% or greater drop in recorded VWRA from one day to the next at any time during the baseline period. Data from 9 mice were excluded for these reasons.

## 2.5. Spontaneous arenas

The mice housed in running-wheel cages were also tested in one of two spontaneous arena tests (no mouse was tested in both). All experiments in arenas (Open Field or Y-maze) were conducted between 1 p.m. (ZT7) and 4 p.m. (ZT10). There was only a single trial in a single arena per mouse. An overhead camera recorded all trials. Distance and location information were calculated by ANY-maze software (Stoelting Co., Wood Dale, IL) and calculation of all outcome measures was automated. Illumination during the experiment was kept as low as possible while still allowing accurate tracking. The arenas were thoroughly wiped with 70% ethanol before each trial. The order of tests was balanced so that the average time of day was approximately the same between groups. The same experimenter handled the mice on the day of the test for all tests.

**2.5.1. Y-maze.** The Y-maze (Med Associates, Vermont, USA) was an arena with three arms (36 cm long); one arm had an additional 14 cm-long entry compartment that was separated from the rest of the maze by a removable door. Mice were placed in the entry compartment, the door was removed, and the mouse was then allowed to explore the Y-maze for 5 minutes. Scoring of arm entries was automated from the location data calculated by the software; a mouse had to be in an arm for at least one uninterrupted second for it to be considered an entry. Each new arm entry was defined as a spontaneous alternation if the two previous entries were into the two other maze arms, and the rate of spontaneous alternation was quantified as the number of spontaneous alternations divided by the total number of arm entries after the first. The test was administered on 28 mice; one mouse was removed from analysis due to making only one arm entry.

**2.5.2. Open field.** A mouse was placed in the center of an opaque white open field arena (45 x 45 cm) and allowed to explore for 30 minutes. A mouse was considered in the "center" of the arena if its center point was more than 10 cm from any of the walls. The test was administered on 45 mice.

## 2.6. Reversal learning

Mice were housed in Phenotyper cages (Noldus, Wageningen, The Netherlands), where they were recorded 24 hours per day by a camera mounted on the top of each cage. A food pellet dispenser protruded from one corner of the cage and dispensed 20 mg grain-based dustless precision pellets (Bio-serv, Flemington, NJ). Noldus Ethovision software identified nose pokes from the video, controlled the pellet dispensers, and calculated total distances travelled by the mice. The cages had a small, rectangular shelter in the corner of the cage opposite the food dispenser where the mice typically slept.

The procedure is modified from a study published by Remmelink et al [22]. During "acclimation" (days -5 through -3) and during the irradiation procedure (days 0–2), the Phenotyper cages had a food tray and no cognition wall. For "training" (days -2 and -1; also 3 and 4) the food tray was replaced by a three-holed cognition wall, situated so that mice would need to poke their nose through a hole to access food pellets. Pellets were dispensed on a fixed-ratio 5 schedule, where a pellet was dispensed after every fifth nose poke into the "correct" (left) hole. The pokes did not need to be consecutive. Reversal learning, where the "correct" hole was changed to the hole on the right, began on day 5 and continued through day 8.

The pellet dispensers were tested once per day to ensure proper function, which provided the mice one free food pellet each day. If at any point a mouse lost 10% or more of its body weight and consumed fewer than 70 food pellets, it was given enough free food pellets to bring its total up to 70 food pellets. The study started with 24 mice, but three were excluded: one did not eat, the second was due to a software error, and the third because it slept in a nose-poke position, causing an accumulation of uneaten pellets.

## 2.7. BDNF Western blot

Mice were anesthetized with a mixture of ketamine (120 mg/kg) and xylazine (20 mg/kg), euthanized by exsanguination, and decapitated. Whole extracted brains were flash-frozen in liquid nitrogen. Lysates were made in the same manner as previously reported [23]. The samples were denatured at 100˚C for 5 minutes and loaded onto 4–20% Mini-PROTEAN® TGX™ Precast Gels (Bio-Rad, Hercules, CA, USA). The gels were run at 100 volts and transferred to polyvinylidene difluoride membranes with Trans-Blot Turbo Transfer System (Bio-Rad, Hercules, CA, USA). Membranes were hydrated with a methanol rinse followed by a 1-hour blocking in 5% Non-Fat Dry Milk Omniblock (AB10109-00100; AmericanBio, Inc., Canton, MA, USA) blocking buffer solution in phosphate-buffered saline with 0.1% Tween (PBST). Membranes were probed with primary antibody (anti-BDNF, 1:2,000 in blocking buffer, cat. no: ab108319; Abcam, Cambridge, UK) overnight at 4˚C. After washing with PBST, the membranes were incubated with the secondary antibody (anti-rabbit IgG HRP, 1:5,000 in blocking buffer, cat. no: NA934, GE Healthcare, Chicago, IL, USA) for 1 hour at room temperature. After imaging, membranes were re-probed with a primary antibody against GAPDH (anti-rabbit, 1:1,000, cat. no: ab9485; Abcam, Cambridge, UK) as a loading control. Immunoreactive complexes were visualized using Super Signal West Pico Chemiluminescent Substrate (Thermo Fisher Scientific, Waltham, MA, USA), imaged with the ChemiDoc MP Imaging Systems Image Lab 6.0.1 (Bio-Rad, Hercules, CA, USA), and densitometry data was quantified with ImageJ.

## 2.8. Statistical analysis

Distance and pellet data were exported from Ethovision for analysis. Custom python code was used to format and plot the data and conduct statistical tests. Data were assessed for normality using a Shapiro-Wilks normality test rejecting normality at $p < 0.05$. A Mann-Whitney rank test was used when normality was rejected, otherwise independent two-tailed t-tests were used with an alpha level of 0.05. Effect sizes are reported as Cohen's d. Correlations are reported as Pearson's *r*. Line plots show the mean values with error bars or shaded areas showing the standard error of the mean. Box plots use matplotlib defaults, displaying the medians (center line), interquartile range (box), and the most extreme values within 1.5 times the interquartile range (whiskers). In all figures, red lines/dots/shading represent the Irrad group, and blue lines/dots/shading represent the Sham group.

# 3. Results

## 3.1. Arena tests

The timeline for running wheel and arena tests (Y-maze or open field) are shown in Fig 1A. Mice were randomized into two groups: Irrad, which underwent either three days of a targeted pelvic irradiation procedure, or Sham, which underwent three days of an identical procedure except the mice were left outside of the irradiator. On the third day after completing the irradiation procedure, each of these mice were also tested in one of two spontaneous arena tests: either a 5-minute Y-maze test or a 30-minute open field test.

Similar to previously published work [21], we found a large decrease in voluntary wheel-running activity (VWRA) in Irrad mice compared to Sham (Fig 2A). The decrease was large and statistically significant (Fig 2B, d = 1.78, t = 15.64, $p < 10^{-22}$, n = 67). Total distance in the Y-maze was lower in Irrad animals, though the difference was not statistically significant (S2A Fig, d = 0.72, t = 1.92, p = 0.0663, n = 27). Spontaneous alternation rates were significantly lower in the Irrad animals (Fig 2C, d = 0.85, t = 2.34, p = 0.0279, n = 27), suggesting cognitive impacts of irradiation that may be related to spatial memory. Total distance in the open field test, similar to the Y-maze, was slightly but not significantly lower in Irrad animals (S2C Fig, d = 0.16, t = 0.54, p = 0.5935, n = 45). There was no significant difference in center time, a common measure of anxiety, between Irrad and Sham animals (S2D Fig, d = 0.35, t = 1.17, p = 0.2472, n = 45).

Distances in either of the two arenas showed little if any correlation with total VWRA over the three post-irradiation days (S2E & S2F Fig), suggesting that 5–30 minute arena tests are not an effective way to measure the fatigue induced by irradiation. VWRA showed a significant correlation with spontaneous alternation in the Y-maze (Fig 2D, r = 0.49, p = 0.017, n = 23), suggesting that the fatigue-like and cognitive behavioral changes may share common mechanisms. Unsurprisingly, there was no correlation between VWRA and center time in the open field test (S2G Fig).

## 3.2. Reversal learning

It has been previously demonstrated that in a prostate cancer population, patients experiencing fatigue after radiation therapy take longer to perform a cognitive task but do not show lower levels of performance accuracy [5]. To test whether a similar phenomenon existed in mice, we sought a cognitive test that was not time-restricted in the way that the arena tests were. We chose to test reversal learning in an automated 24-hour video recording setup. Mice were first trained to nose-poke into the left hole of a three-holed wall to receive food; the third day after completing the irradiation procedure, the rewarded hole was changed to the one on the right.

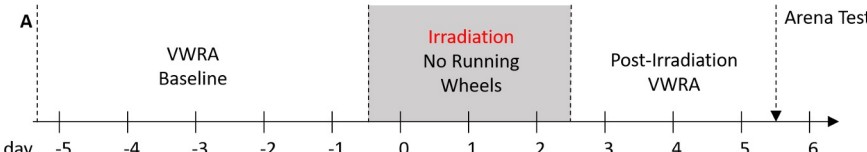

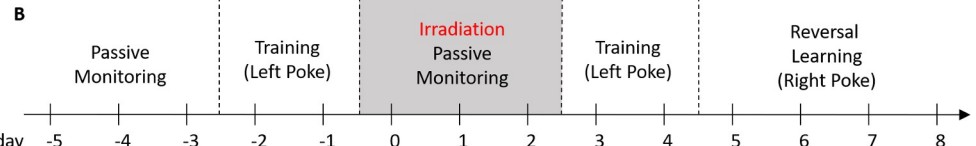

**Fig 1. Experiment design. (A)** Arena tests (n = 76 mice). Day -5: Baseline VWRA for 5 days. Day 0: Mice were irradiated for 3 days while housed in cages without running wheels. Day 3: Post-irradiation VWRA for 3 days. Day 6: arena test (open field or Y-maze, not both). **(B)** Reversal learning (n = 24 mice). Days -5 through -3: Mice acclimated to the cage without a cognition wall for three days. Day -2: A cognition wall was inserted for training, where left hole nose pokes were paired with food. Day 0: The cognition wall was removed for three days of irradiation. Day 3: The cognition wall was reinserted into the cage for further training. Day 5: Reversal learning, where right hole nose pokes were paired with food.

Our primary outcome measure was the percentage of nose-pokes into the correct hole after it was switched for reversal learning. The mice could participate in the task as little or as much as they wanted and receive food rewards as quickly or as slowly as they wanted. The timeline of the experiment is shown in Fig 1B.

Video monitoring allowed us to record total distance travelled in the home cage, and again similar to published work on cancer-related fatigue, we found that locomotor activity was much lower the Irrad group than Sham (Fig 3A). The effect of irradiation on the total distance travelled was statistically significant (Fig 3B, d = 1.62, t = 6.17, $p < 10^{-5}$, n = 21), which shows that we can still measure fatigue-like behavior while the mouse is feeding via the cognition wall. There was also a significant effect of irradiation on total food consumed (Fig 3C, d = 1.46, t = 4.71, $p < 10^{-3}$, n = 21). There was a strong and statistically significant correlation between food consumed and distance travelled in the Irrad group (Fig 3D, r = 0.76, p = 0.006) but not the Sham group (r = 0.21, p = 0.563), suggesting that fatigue and reduced feeding may be related.

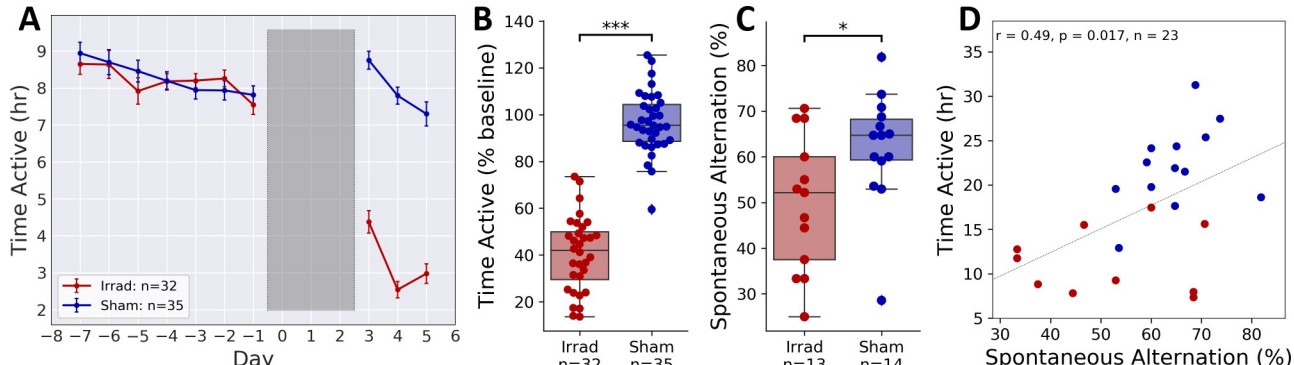

**Fig 2. Arena tests. (A)** Mean daily VWRA totals. Grey shaded area indicates days of irradiation. **(B)** VWRA across the 3-day post-irradiation period normalized to the individual 7-day baseline daily means. **(C)** Spontaneous alternation rate in the Y-maze test. **(D)** Correlation between VWRA and spontaneous alternation in the Y-maze. $^{*}p < 0.05$, $^{***}p < 0.0005$.

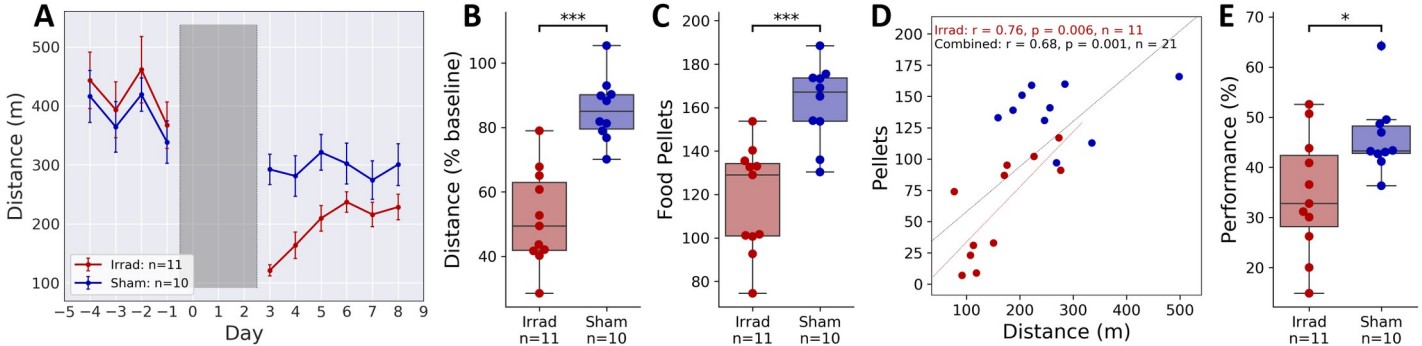

**Fig 3. Reversal learning.** "Performance" is the percentage of nose-pokes into the rewarded hole of the cognition wall. Performance line plots (E, F, H, I) display the mean performance over a 30 nose-poke rolling window. (**A**) Mean daily distances traveled. Grey shaded area indicates the three days of irradiation. (**B**) Distances travelled across the 6-day post-irradiation period normalized to the total on the day before irradiation. (**C**) Food pellets consumed per day across the 6-day post-irradiation period. (**D**) Correlation between VWRA and food consumption for each group and for the combined data. (**E**) Mean performance cumulative over the first night of reversal learning. $^*p < 0.05$, $^{***}p < 0.0005$.

For two days after irradiation, mice continued with training and showed stable performance with little difference between groups (S5B & S5E Fig). To evaluate the subsequent reversal learning, we first calculated the performance (percent of pokes into the correct hole) as a rolling average of 30 pokes. When looking at performance with respect to time, we found no difference between groups in the initial training before irradiation (S5A Fig) but a clear difference during reversal learning after irradiation (S5C Fig). We next calculated performance over the entire first night of data collection. We chose to look at the first night because most locomotor activity and food dispensing took place at night (S4 Fig) and most of the learning (performance change) took place over the first night (S5C Fig). Comparing cumulative performance over the first night of reversal learning relative to training, the difference between groups is statistically significant (Fig 3E, d = 1.02, t = 2.65, p = 0.0167, n = 21). When looking at performance with respect to the number of pokes, we find that performance during training is again the same between groups (S5D Fig), but that the changes in reversal learning disappear (S5F & S6A Fig, d = 0.11, t = 0.24, p = 0.815, n = 21). It is clear that mice in the Irrad group are taking more time to make the equivalent number of nose pokes, suggesting that mice with irradiation-induced fatigue learn slower with respect to time primarily because they engage with a task at a slower pace. However, they do not learn slower with respect to engagement with the task.

## 3.3. BDNF

We next hypothesized that changes in BDNF expression in the brain may be associated with fatigue-like and cognitive behaviors. We measured both proBDNF and mBDNF in whole brain lysates from the mice described in Fig 2, which were housed with running wheels and underwent one of two arena tests. We used Western blot analysis to measure levels of mBDNF and proBDNF normalized to GAPDH (representative bands were shown Fig 4A; all raw Western Blot images used in densitometric analysis are included in S1 Raw images), and we found a significant effect of irradiation on mBDNF levels (Fig 4B, d = 0.48, U = 260.00, p = 0.0334, n = 56) but not on proBDNF (S7A Fig, d = 0.03, U = 353, p = 0.5487, n = 56). Since mBDNF expression may be upregulated by exercise, and irradiated mice exercised less on their running wheels (Fig 2A), we tested whether the change in mBDNF levels correlated with VWRA. Surprisingly, we found very little correlation (S7B Fig), which suggests that other factors, such as the irradiation process itself, may be causing the change in mBDNF levels.

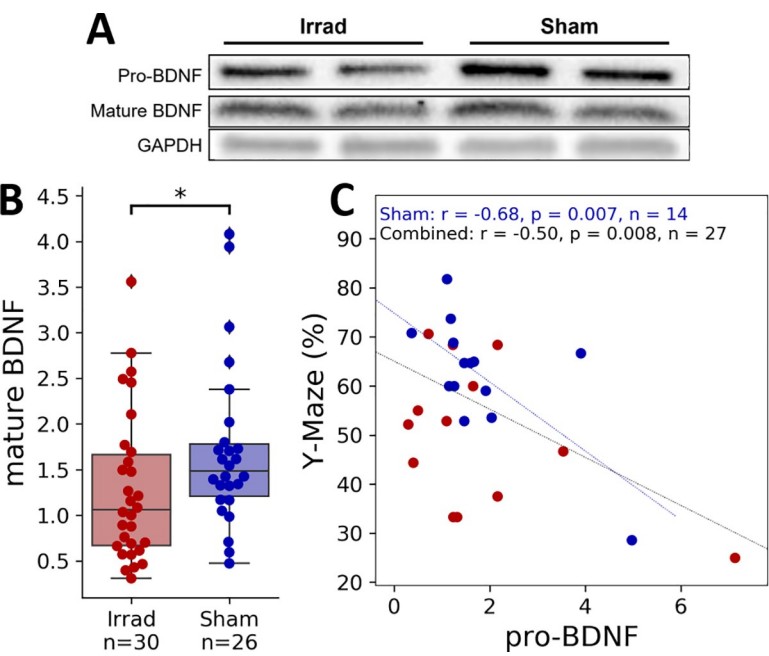

**Fig 4. BDNF levels.** BDNF levels were measured in mouse whole brain lysates via Western Blot analysis three days after irradiation and one day after Y-maze or open field testing. (**A**) Representative bands from a Western blot for mBDNF, proBDNF, and GAPDH (all raw Western blot images used for densitometric analysis were included in S6 Fig). (**B**) Densitometric analysis results of mBDNF normalized the loading control, GAPDH. (**C**) Correlation between measured proBDNF levels and spontaneous alternation in the Y-maze. $^*p < 0.05$.

Even though irradiation did not change proBDNF levels, we did find a significant negative correlation between proBDNF and spontaneous alternation performance in the Y-maze (Fig 4C, r = -0.50, p = 0.008, n = 35). This was not the case for mBDNF (S7D Fig), which suggests increased levels of proBDNF may negatively affect cognition in a way that is not related to the irradiation. This interpretation is supported by a significant correlation only in the Sham group (r = -0.86, p = 0.007, n = 14) but not the irradiated group (r = -0.49, p = 0.093, n = 19). Further, the overall correlation may be highly driven by an irradiated mouse with an unusually high level of proBDNF; when calculating the Spearman correlation, which can be more robust to outliers, correlation within the Sham group is still statistically significant (r = -0.60, p = 0.024) but the overall correlation is no longer significant (r = -0.29, p = 0.137). Open time in the open field did not significantly correlate with levels of mBDNF or proBDNF (S7E & S7F Fig).

## 4. Discussion

In previous studies [21], we have used a mouse model of radiation therapy for treatment of prostate cancer to show that pelvic irradiation alone is sufficient to induce a profound fatigue-like behavior that can be measured as a decline in VWRA or home cage locomotor activity. In this study, we found that this pelvic irradiation also induces a change in cognitive behavior, most notably as a decrease in spontaneous alternation in a Y-maze. We also found that the behavioral changes after irradiation are accompanied by lower levels of mBDNF in the brain.

The mechanism by which irradiation constrained to a pelvic region affects both cognitive and fatigue behaviors is unknown. In a previous study we established that weight loss is also a result of this irradiation procedure [21], so a possibility that immediately occurred to us is that reduced food consumption, as shown in Fig 3C, induces these changes. However, this may be

unlikely because food restriction in rodents typically results in an increase in VWRA [24]. We would also anticipate food restriction to, if anything, show an increase in spontaneous alternation behavior, as food restriction has been shown to benefit spatial learning and memory in rodents [25] and increase BDNF and plasticity in the hippocampus [12], both of which are thought to be important for spontaneous alternation behavior [26]. A more plausible cause of behavioral changes is a systemic inflammatory response, as radiation can elevate circulating levels of pro-inflammatory cytokines [27] that may in turn alter behavior. This may resemble sickness behavior, which can include both fatigue-like behavior and cognitive deficits [28].

Exercise has been linked with spatial learning and memory as well as hippocampal neurogenesis [29], and also with an increase in hippocampal mBDNF expression [11, 30]. We initially suspected that lower levels of BDNF in the Irrad group are caused by the decrease in voluntary exercise on the running wheels. However, if this were the case, we would expect to see a correlation between VWRA and mBDNF levels, which we surprisingly did not see (S7B Fig). A future study could help clarify this relationship by using control groups of mice that are not able to exercise on running wheels. Levels of mBDNF also did not significantly correlate with spontaneous alternation rates, which is again somewhat surprising, since there is evidence that mBDNF in the hippocampus [31] and prefrontal cortex [32] is associated with improvements in spatial memory tests. However, spontaneous alternation did negatively correlate with proBDNF levels, and it has been shown previously that proBDNF in the hippocampus was associated with cognitive impairment [8]. In our study it is possible that irradiation induces signals to the CNS that shift the balance between the pro-survival and pro-apoptotic properties of mBDNF and proBDNF, respectively, which can impair cognitive abilities.

We found a negative correlation between proBDNF levels and spontaneous alternation (Fig 4C), but this result may be difficult to reconcile with other results. First, spontaneous alternation also correlated with VWRA (Fig 2D), but we did not find any correlation between proBDNF and VWRA (S7C Fig). Additionally, irradiation affected VWRA and spontaneous alternation, but did not affect proBDNF levels. It may be explained by proBDNF having an effect on spontaneous alternation that is present in the Sham group and masked by the irradiation procedure, which our results would suggest. Thus, the correlation between VWRA and spontaneous alternation is primarily driven by group differences that are not related to proBDNF. Future experiments that focus on the relationship between proBDNF and spontaneous alternation behavior may help clear this up.

Another future experiment that may be critical to elucidating the role of BDNF in radiation-induced changes in behavior would be to measure BDNF expression in different brain regions. There are reports of opposite effects of BDNF in depression-like behaviors depending on brain region, where increased BDNF in the hippocampus and frontal cortex can have antidepressant-like effects and BDNF in the ventral striatum can produce depression-like behaviors [33]. In contrast, there is evidence that BDNF in the ventral striatum can produce cognitive improvements [34], though there are not many reports of this in the literature. Since fatigue behaviors can be closely related to problems with cognition and depression, these symptoms may share distinct mechanisms related to BDNF expression.

Another important result of this study is that post-irradiation performance in the reversal learning task is impaired when evaluated with respect to time but not with respect to engagement in the task. This resembles some results seen in human subjects; for example, deficits in speed but not accuracy at cognitive tasks have been shown in CRF [5] and CFS [7] patients. We believe the most likely explanation for our results is that the willingness to engage in the task is compromised by irradiation, but the reversal learning itself is not. This would suggest that the circuitry underlying reversal learning, such as the orbitofrontal cortex and striatum

[35], may be less affected by the pelvic irradiation than the circuitry associated with spontaneous alternation, for example connections between the hippocampus and temporal cortex [26].

Our understanding of the reversal learning results would benefit from future experiments that could expand on these results to distinguish the learning from related behaviors, such as attention and response times using a serial reaction time test, or motivation using a progressive ratio test. Both of those examples could be used with a similar setup to test rates versus accuracy of goal-directed behaviors. Combining these with brain-circuit-specific measures could help understand how behaviors or symptoms related to fatigue may be dissociable or overlapping.

Additionally, our reversal learning results are an important demonstration that a study where task participation is constrained by time could erroneously show cognitive deficits in fatigued participants, whether mice or human. Had our experiment design been to measure performance over one night of reversal learning, we would have reached the opposite conclusion, that there was an effect of pelvic irradiation on reversal learning. Therefore, when designing studies in which fatigue may be a factor, measurements of learning performance or accuracy should be completely independent from speed.

In conclusion, we found that in addition to inducing fatigue-like behavior, pelvic irradiation can induce impairments in cognitive behaviors, which may resemble a symptom cluster observed in humans. We also found in a reversal learning task that it takes more time for the fatigued mice to learn, but that there is no decrease in overall task performance. Finally, we find that the brains of fatigued mice show lower levels of BDNF after irradiation, and that elevation of brain proBDNF correlates with decreased cognitive performance, which suggests that changes in BDNF levels may be affecting behaviors following irradiation.

## Supporting information

**S1 Fig. Circadian VWRA.** Mean VWRA time active for each minute of recording, with darker colors representing more activity, light colors representing lower levels of activity, and white representing no data. Irradiation took place on days 0, 1, and 2. Zeitgeber time is the number of hours after lights are turned on at 6 a.m.
(TIF)

**S2 Fig. Spontaneous arena behaviors. (A)** Distance travelled in the 5-minute Y-maze test. **(B)** The total number of arm entries during the Y-maze were not significantly affected by irradiation (d = 0.60, t = 1.56, $p$ = 0.132, n = 27). **(C)** Distance travelled in the 30-minute open field test. **(D)** Center time in the open field test. **(E)** There were no significant correlations between VWRA and distances travelled in the Y-maze. **(F–G)** There were no significant correlations between VWRA and distances travelled (F) or open time (G) in the open field.
(TIF)

**S3 Fig. Arena performance over time.** Since irradiation showed different effects on reversal learning over time vs. over participation with the task, we did a similar analysis on the arena behaviors. **(A)** Plotting spontaneous alternation behavior over time in the Y-maze, distance travelled, or total arm entries had little effect on the appearance of the plots. **(B)** Plotting center time in the open field over time in the arena or over distance traveled had little effect on the appearance of the plots. The effect size is plotted with a green line without shading and uses the right-hand axis labels.
(TIF)

**S4 Fig. Circadian activity during reversal learning. (A–B)** Mean locomotor distance totals for each minute of recording, with darker colors representing greater distances, light colors

representing lower lesser distances, and white representing no data. **(C–D)** Mean number of food pellets dispensed during each minute of recording, with darker colors representing more pellets, light colors representing fewer pellets, and white representing no data. Irradiation took place on days 0, 1, and 2, and no food pellets were dispensed during this time (mice had *ad libitum* access to chow). Zeitgeber time is the number of hours after lights are turned on at 6 a.m.
(TIF)

**S5 Fig. Reversal learning behavior. (A–C)** Performance over time during training (A), retraining (B), and reversal learning (C). Dark shading represents the dark cycle (night). **(D-F)** Performance over the total number of pokes during training (D), retraining (E), and reversal learning (F).
(TIF)

**S6 Fig. Reversal learning statistics. (A)** Mean performance cumulative over the first 1536 pokes of reversal learning, which is the mean number of nose-pokes over the first night across all mice in the Sham group. **(B)** There were no significant correlations between distance traveled and performance during the first night of reversal learning. **(C)** There were large significant correlations between food pellets dispensed and performance during the first night of reversal learning, particularly for the Irrad group. This is not surprising, as performance at the task causes food pellets to dispense.
(TIF)

**S7 Fig. BDNF levels and behavior. (A)** Densitometric analysis results of proBDNF normalized to GAPDH. **(B–C)** There were no significant correlations between mBDNF or proBDNF and VWRA. **(D)** There were no significant correlations between mBDNF and spontaneous alternation in the Y-maze. **(E–F)** There were no significant correlations between mBDNF or proBDNF and open time in the open field test.
(TIF)

**S1 Raw images. BDNF Western blot images.** Unedited raw images from each western blot used in densitometric analyses shown in Figs 4 and S7.
(PDF)

## Acknowledgments

We would like to thank the NHLBI Murine Phenotyping Core for their invaluable help with experiments, including Dr. Danielle Springer, Michele Allen, Audrey Noguchi, Heather Potts, and Morteza Pieravi. We would also like to thank the NHLBI Animal Surgery and Resources Core for help with taking blood and tissue samples.

## Author Contributions

**Conceptualization:** Brian S. Wolff, Li Rebekah Feng, Leorey N. Saligan.

**Formal analysis:** Brian S. Wolff.

**Investigation:** Sumiyya A. Raheem, Sarah A. Alshawi, Jeniece M. Regan.

**Methodology:** Sumiyya A. Raheem, Sarah A. Alshawi, Jeniece M. Regan, Li Rebekah Feng, Leorey N. Saligan.

**Project administration:** Leorey N. Saligan.

**Resources:** Leorey N. Saligan.

**Software:** Brian S. Wolff.

**Supervision:** Brian S. Wolff, Li Rebekah Feng.

**Writing – original draft:** Brian S. Wolff.

**Writing – review & editing:** Sumiyya A. Raheem, Sarah A. Alshawi, Jeniece M. Regan, Li Rebekah Feng, Leorey N. Saligan.

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
