## [Decision Letter · Decision Letter 0]

30 Apr 2020

PONE-D-20-08431

Induction of fatigue-like behavior by pelvic irradiation of male mice alters cognitive behaviors and BDNF expression.

PLOS ONE

Dear Dr. Saligan,

Thank you for submitting your manuscript to PLOS ONE. After careful consideration, we feel that it has merit but does not fully meet PLOS ONE’s publication criteria as it currently stands. Therefore, we invite you to submit a revised version of the manuscript that addresses the points raised during the review process.

You can see that the Reviewers found your manuscript to be good but there are further experiments and analysis that needs ot be done to help with clarification. You need to follow all of their suggestions and then revise the manuscript accordingly. This revised manuscript will be sent back to the Reviewers to determine whether their criticisms have been sufficiently answered.

We would appreciate receiving your revised manuscript by Jun 14 2020 11:59PM. To enhance the reproducibility of your results, we recommend that if applicable you deposit your laboratory protocols in protocols.io, where a protocol can be assigned its own identifier (DOI) such that it can be cited independently in the future. For instructions see: http://journals.plos.org/plosone/s/submission-guidelines#loc-laboratory-protocols

We look forward to receiving your revised manuscript.

Kind regards,

Michelle M. Adams, Ph.D.

Academic Editor

PLOS ONE

Journal Requirements:

Reviewers' comments:

Reviewer's Responses to Questions

**Comments to the Author**

1. Is the manuscript technically sound, and do the data support the conclusions?

Reviewer #1: Partly

Reviewer #2: Yes

Reviewer #3: Partly

Reviewer #4: Partly

2. Has the statistical analysis been performed appropriately and rigorously? 

Reviewer #1: Yes

Reviewer #2: Yes

Reviewer #3: Yes

Reviewer #4: Yes

3. Have the authors made all data underlying the findings in their manuscript fully available?

Reviewer #1: No

Reviewer #2: Yes

Reviewer #3: Yes

Reviewer #4: Yes

4. Is the manuscript presented in an intelligible fashion and written in standard English?

Reviewer #1: Yes

Reviewer #2: Yes

Reviewer #3: Yes

Reviewer #4: Yes

5. Review Comments to the Author

Reviewer #1: The aim of this study is to clarify whether there is connection between fatique-like behavior and cognitive behavioral changes in mice and compare BDNF levels in brain.

There are several major issues with this current paper, regarding organization, data interpretation and visualization.

It is not clear why researchers used normal mice and targeted them with pelvic irradiation instead of using prostate cancer mouse model which is commercially available. If prostate cancer patients receiving radiation treatment have specific cognitive deficits with cancer-related fatigue and this needs to be investigated, the reason to use normal mice to show relationship between BDNF and CRF should be explained. Prostate cancer mouse model receiving pelvic irradiation and normal mouse received irradiation probably will give different response.

Although general structure of the manuscript is just fine, some issues must be revisited

Page2, Line 51-57, entire paragraf should be rewritten and organized.

In the entire manuscript there is no definition of BDNF. Appreviation is not described.

There is no data about weight measurement in the manuscript. During the experimention, weight loss may contribute to the behavioral assesments.

Page 3, Line 89: Is there any data for elevated 0-maze as mentioned?

Page 4, Line 132 and page 7, Line 254: There is no data for WB provided in my copy.

Page7, Line 261: It is not clear whether previous study is published or not , no reference was given. May be those data should be included in this manuscript.

Without Wb data, it is difficult to conclude the relationship BDNF level, irradiation and cognitive performance.

Reviewer #2: This is an interesting and timely study. The following issues should be addressed:

1. Please describe how shielding was performed for the localized radiation exposure. Please also indicate the details with regard to the irradiator used.

2. “Data from 8 mice were excluded from VWRA analysis due a 50% or greater drop in recorded VWRA totals during the baseline recording period.” It is not clear what this means; please explain.

3. Please refer to the time line to indicate the sequence of the behavioral tests and the interval between them in the methods section.

4. Please do not use the word “sacrificed” as standard for most journals now.

5. There are limitations to analyze whole brain levels of BDNF as more regional changes would be anticipated to play a role in wheel running and performance in the behavioral tests used in this study.

6. The reduced wheel running activity could affect BDNF levels in these animals; so what is a consequence of radiation and what is a consequence of reduced running seems hard to distinguish without a locked wheel control. This should be acknowledged and discussed.

7. It would be good to analyze whether wheel running in the light and/or dark periods was affected and whether there were circadian changes.

8. There seems no need to show the data as a percent of baseline with these striking group differences; please show actual data that is not normalized as well.

9. It seems better to only show the significant correlations and include the other data as supplementary table or figure panels.

10. The correlation shown in Figure 4F does not seem very convincing and might be driven by some individual data points.

Reviewer #3: The authors describe the effect of pelvic irradiation of male mice on cognitive behaviors and whole brain BDNF contents.

The paper showed that pelvic irradiation produced a reduced rate of spontaneous alteration in Y maze test and expression changes in mature/pro BDNF levels in whole brain. Although this study contains interesting information, mechanistic information may be less focused.

1)Why did authors examine protein BDNF levels in whole brain?

It has been reported that central BDNF exhibits opposite actions depending on the site of actions. For example, in preclinical studies, inflammation promotes reduced BDNF in the PFC and hippocampus, as well as increased BDNF in the NAc of the brain, resulting in a

depression-like phenotype in rodents (Zhang et al. Curr Neuropharmacol. 2016;14(7):721-31.)

If they try to suggest the mechanistic correlation between cognitive/emotional function and BDNF expression, it is strongly recommended to examine the BDNF levels in at least either hippocampus, PFC, or NAc in the pelvic irradiation model. Or they could additionally examine inflammatory status (ex. cytokines, microglia activation) in whole brain levels as they discussed in the Discussion (Page 7, L274: A more plausible cause of behavioral changes is a systemic inflammatory response…).

2)The Fig. 4D, 4F, 4H seems to show pro BDNF in irradiation animals. Why are the expression plots of proBDNF in each Figure so different? (The animals number could be 4D = 4F + 4H)

-It is recommended to include both irradiation (closed circle) and sham (open circle) animals in correlation plots.

-Please show the number of animals in each Figure.

3) In Fig 2I, VWRA (active time) correlates with Y maze's spotaneous alterlation. In Fig.4F, proBDNF is inversely correlated with Y maze's spotaneous alterlation. So why does VWRA not correlate with proBDNF in Fig4D? Please discuss this point.

Reviewer #4: This manuscript describes the assessment of fatigue and cognitive functions in a mouse model of pelvic irradiation. The premise of the study stems from findings in cancer patients, especially from prostate cancer patients, that fatigue is a common finding following radiation therapy and that there may or may not be an associated cognitive decline. Voluntary running wheel exercise was used before and after irradiation to assess the extent of fatigue resulting from irradiation. The mice were also assessed for working spatial memory, anxiety, reversal learning, and tissue levels of proBDNF and mBDNF in the brain. The authors observed a significant level of fatigue in the irradiated cohort which was supported by a significant reduction in voluntary wheel running activities following irradiation. Associated with the reduced voluntary activity was an impaired performance in the spontaneous alteration behavior in the Y maze exploration. Most interesting was the outcome in the reversal learning test. The authors showed that, without constrained by time, the irradiated mice were able to learn and perform to the same level of accuracy as sham controls in obtaining food pellets in the reversal phase of the nose pole/food reward learning paradigm. Consistent with the voluntary wheel running outcome, irradiated mice showed a significant reduction in ambulatory activities in the reversal learning arena, and the reduced activity correlated with reduced food intake. The authors also observed a significant reduction of mBDNF in the brains of irradiated mice, which may be a result of reduced overall physical activity. The studies are carefully designed and the results are relevant to a significant problem in cancer patients. The study results should be of interest to clinician and investigators in the radiation and cancer therapy field in general. Several issues warrant further clarification from the authors.

1. Given that importance of the results from reversal learning, it would have been nice to have a separate test to confirm the results. In the absence of such confirmation, authors should discuss what future studies may be employed to strengthen the conclusion in the Discussion.

2. It was not clear initially that two separate sets of mice were used for the voluntary wheel running/OF/Y maze and the reversal learning and that mice in the voluntary wheel running group only went through OF or Y maze, but not both arena tests. Providing these information earlier in the Methods section would have been helpful.

3. Elevated zero maze was not performed (or at least the data were not provided), but was mentioned in line 89.

4. There was a mention OF and Y maze tests were performed between ZT7 and ZT10 (line 89-90). Please clarify what is ZT7 and ZT10.

5. The reversal learning started on Day 5 (or 3 days after irradiation), but it was not clear how many days were devoted to the reversal phase. Was there a proficiency criterion to reach before the test was terminated?

6. As a related question – there were two more days of training after irradiation. Were the performance from the irradiated mice comparable to the sham controls, and how was the performance compared to the pre-irradiation phase?

7. To measure performance without time constraint, the authored chose to analyze performance during the first 1600 nose poke in the reversal phase. What was the rationale for selecting 1600 nose pokes as the cutoff?

8. It will be helpful if the authors can indicate on the timescale (Figures E and F) which segments are the dark phase and which are the light phase. Likewise, because the time required to complete 1600 nose pokes in the reversal phase is different between sham and irradiated animals, it will be helpful for the authors to provide that information.

9. Figures 3G and 3J are a percentage over a percentage (performance during the reversal phase over performance during the first night of training). It is also not clear if 3J uses the average performance over the entire reversal phase, or just during the first 1600 nose pokes, and normalizes it to the first 1371 nose pokes. It may be helpful in visualizing the data if the authors can plot out the performance during the first night or the first 1600 nose pokes of the reversal learning and performance of the first night or the first 1371 nose pokes, respectively, during the training phase for side-by-side comparisons.

10. The authors put sample sizes in 2A and 3A, but did not provide the same information for Y maze, OF, and BDNF plots. For consistency, these information should be provided.

6. PLOS authors have the option to publish the peer review history of their article (what does this mean?). If published, this will include your full peer review and any attached files.

Reviewer #1: No

Reviewer #2: No

Reviewer #3: No

Reviewer #4: Yes: Ting Ting Huang

---

## [Author Response · Author response to Decision Letter 0]

29 May 2020

Reviewer #1

The aim of this study is to clarify whether there is connection between fatique-like behavior and cognitive behavioral changes in mice and compare BDNF levels in brain.

There are several major issues with this current paper, regarding organization, data interpretation and visualization.

1. It is not clear why researchers used normal mice and targeted them with pelvic irradiation instead of using prostate cancer mouse model which is commercially available. If prostate cancer patients receiving radiation treatment have specific cognitive deficits with cancer-related fatigue and this needs to be investigated, the reason to use normal mice to show relationship between BDNF and CRF should be explained. Prostate cancer mouse model receiving pelvic irradiation and normal mouse received irradiation probably will give different response.

Authors: This is an important point for us to address, and we thank you for raising it. We used a normal mouse model instead of a prostate cancer mouse model to investigate the behavioral effects specific to irradiation. It is not clear from clinical populations what fatigue-like effects are due to cancer, what effects are due to treatment, and what effects are caused by the interaction of the two. We have added this information into the Introduction section.

2. Although general structure of the manuscript is just fine, some issues must be revisited

Page2, Line 51-57, entire paragraf should be rewritten and organized.

Authors: We have reworded this paragraph to make it clearer. 

3. In the entire manuscript there is no definition of BDNF. Appreviation is not described.

Authors: We have defined the abbreviation of BDNF the first time it is used in the introduction.

4. There is no data about weight measurement in the manuscript. During the experimention, weight loss may contribute to the behavioral assesments.

Authors: This is correct and an important point. We did not include data on body weight in this manuscript. Weight loss occur after this irradiation procedure, as described in previous work (Wolff et al., Scientific Reports, 2018) cited in the manuscript. This weight loss may contribute to changes in behaviors, and we discussed food consumption in the second paragraph of our original discussion section. In this revised draft, we added a note about weight loss to more thoroughly address this important point.

5. Page 3, Line 89: Is there any data for elevated 0-maze as mentioned?

Authors: We apologize, this was an editing error on our part and the mention of elevated 0-maze is removed.

6. Page 4, Line 132 and page 7, Line 254: There is no data for WB provided in my copy.

Authors: This was an unfortunate omission, thank you for pointing it out. Western blot data have been added to Figure 4 (representative bands) and supplemental Fig S6, which includes all raw, unedited Western Blot images used for densitometric analysis.

7. Page7, Line 261: It is not clear whether previous study is published or not, no reference was given. May be those data should be included in this manuscript.

Authors: These data are published, so the reference has been added.

8. Without Wb data, it is difficult to conclude the relationship BDNF level, irradiation and cognitive performance.

Authors: The Western blot data have been added to Figure 4 (representative plot) and supplemental Fig S6, which includes all raw, unedited Western Blot images used for densitometric analysis.

Reviewer #2

This is an interesting and timely study. The following issues should be addressed:

1. Please describe how shielding was performed for the localized radiation exposure. Please also indicate the details with regard to the irradiator used.

Authors: We have added the irradiator model into the methods. The shielding is described in detail in Wolff et al., JoVE 2017. Rather than duplicate the published description, we have clarified that shielding details are contained in the cited publication.

2. “Data from 8 mice were excluded from VWRA analysis due a 50% or greater drop in recorded VWRA totals during the baseline recording period.” It is not clear what this means; please explain.

Authors: This has been clarified in the text, and we also fixed an incorrect count (there were actually 9 excluded animals.

3. Please refer to the time line to indicate the sequence of the behavioral tests and the interval between them in the methods section.

Authors: We regret the confusion on this point. Since multiple reviewers mentioned this, we clarified that mice underwent only one of the two behavioral tests in the methods section, results section, and in the legend for Figure 1.

4. Please do not use the word “sacrificed” as standard for most journals now.

Authors: We have changed this to “euthanized”.

5. There are limitations to analyze whole brain levels of BDNF as more regional changes would be anticipated to play a role in wheel running and performance in the behavioral tests used in this study.

Authors: We agree with this and have added a paragraph to the discussion to discuss regional changes.

6. The reduced wheel running activity could affect BDNF levels in these animals; so what is a consequence of radiation and what is a consequence of reduced running seems hard to distinguish without a locked wheel control. This should be acknowledged and discussed.

Authors: This is true, and a very important point that the exercise may be a primary cause of different BDNF levels. We did not see a correlation between wheel running and BDNF levels, but using the control groups you mentioned would be a more convincing experiment. We have edited our discussion of this topic in the third paragraph of the Discussion section to make this point clear.

7. It would be good to analyze whether wheel running in the light and/or dark periods was affected and whether there were circadian changes.

Authors: We did an analysis of circadian changes in VWRA in a previous publication (Wolff et al., Scientific Reports, 2018). We have added a figure (Figure S1) to the supplemental material to show that this study is consistent with earlier results.

8. There seems no need to show the data as a percent of baseline with these striking group differences; please show actual data that is not normalized as well.

Authors: We have changed most of the figures to show non-normalized data. We would agree with the reviewer our absolute measures (e.g. spontaneous alternation or BDNF levels) should be analyzed alongside an absolute measure of physical activity. 

Figures 2B and 3B still show normalized data, as we believe this is the most important measure of “fatigue”. Locomotor activity levels at baseline shows a lot of variability from animal to animal, presumably because some animals prefer to move around more than others. The sources of this variability will likely be present after irradiation as well, meaning we expect the animals that move more before irradiation to also move more after irradiation. We are not interested in this baseline tendency for locomotor activity; what we are interested in is how much this baseline tendency changes. This change in activity (normalized data), rather than the absolute level of activity, is what we consider to be the direct outcome of irradiation. However, figures 2A and 3A display the non-normalized data in order to show that the groups have similar values at baseline.

9. It seems better to only show the significant correlations and include the other data as supplementary table or figure panels.

Authors: Non-significant correlation data have been moved to the supplementary materials.

10. The correlation shown in Figure 4F does not seem very convincing and might be driven by some individual data points.

Authors: It does look like the data point furthest to the right is driving a lot of the correlation. We have noted this in the results section, and we added analysis of the correlation within individual groups. Doing so shows that the correlation only looks robust within the Sham group.

Reviewer #3

The authors describe the effect of pelvic irradiation of male mice on cognitive behaviors and whole brain BDNF contents.

The paper showed that pelvic irradiation produced a reduced rate of spontaneous alteration in Y maze test and expression changes in mature/pro BDNF levels in whole brain. Although this study contains interesting information, mechanistic information may be less focused.

1. Why did authors examine protein BDNF levels in whole brain?

It has been reported that central BDNF exhibits opposite actions depending on the site of actions. For example, in preclinical studies, inflammation promotes reduced BDNF in the PFC and hippocampus, as well as increased BDNF in the NAc of the brain, resulting in a

depression-like phenotype in rodents (Zhang et al. Curr Neuropharmacol. 2016;14(7):721-31.)

If they try to suggest the mechanistic correlation between cognitive/emotional function and BDNF expression, it is strongly recommended to examine the BDNF levels in at least either hippocampus, PFC, or NAc in the pelvic irradiation model. Or they could additionally examine inflammatory status (ex. cytokines, microglia activation) in whole brain levels as they discussed in the Discussion (Page 7, L274: A more plausible cause of behavioral changes is a systemic inflammatory response…).

Authors: Thank you for raising this point. These are very good ideas, and we have added a paragraph to the discussion about this.

2. The Fig. 4D, 4F, 4H seems to show pro BDNF in irradiation animals. Why are the expression plots of proBDNF in each Figure so different? (The animals number could be 4D = 4F + 4H)

-It is recommended to include both irradiation (closed circle) and sham (open circle) animals in correlation plots.

-Please show the number of animals in each Figure.

Authors: We agree this was confusing, and we are now displaying the number of animals in each figure. The Irrad and Sham groups are displayed separately in red and blue, and we have added correlation analysis for both the overall data set and for the individual groups. We thank you for raising this point, as it has really improved the quality of the manuscript. 

The values for pro-BDNF displayed in the figures mentioned are almost all the same, with a few differences due to missing data. Most obviously, there is a pro-BDNF value of about 7 that appears in spontaneous alternation correlation plot (now Fig 4D), but not in the running wheel correlation plot (now Fig S5B). The running wheel data for this mouse were excluded due to a technical malfunction, where the running wheel was still spinning but not recording data. Because the running wheel was still spinning normally, we included this mouse’s Y-Maze and BDNF data. There were missing data points in each data set, which were described in the methods section, and therefore the sample size numbers don’t always match when comparing one data set to another.

Please note that at another reviewer’s request we have moved non-significant correlation plots to the supplementary data, so the Figure labels (e.g. 4A, 4B, etc.) have changed from the previous version.

3. In Fig 2I, VWRA (active time) correlates with Y maze's spotaneous alterlation. In Fig.4F, proBDNF is inversely correlated with Y maze's spotaneous alterlation. So why does VWRA not correlate with proBDNF in Fig4D? Please discuss this point.

Authors: You are correct, and we agree that this is worth discussing. We have added a paragraph to the Discussion addressing this point.

Reviewer #4

This manuscript describes the assessment of fatigue and cognitive functions in a mouse model of pelvic irradiation. The premise of the study stems from findings in cancer patients, especially from prostate cancer patients, that fatigue is a common finding following radiation therapy and that there may or may not be an associated cognitive decline. Voluntary running wheel exercise was used before and after irradiation to assess the extent of fatigue resulting from irradiation. The mice were also assessed for working spatial memory, anxiety, reversal learning, and tissue levels of proBDNF and mBDNF in the brain. The authors observed a significant level of fatigue in the irradiated cohort which was supported by a significant reduction in voluntary wheel running activities following irradiation. Associated with the reduced voluntary activity was an impaired performance in the spontaneous alteration behavior in the Y maze exploration. Most interesting was the outcome in the reversal learning test. The authors showed that, without constrained by time, the irradiated mice were able to learn and perform to the same level of accuracy as sham controls in obtaining food pellets in the reversal phase of the nose pole/food reward learning paradigm. Consistent with the voluntary wheel running outcome, irradiated mice showed a significant reduction in ambulatory activities in the reversal learning arena, and the reduced activity correlated with reduced food intake. The authors also observed a significant reduction of mBDNF in the brains of irradiated mice, which may be a result of reduced overall physical activity. The studies are carefully designed and the results are relevant to a significant problem in cancer patients. The study results should be of interest to clinician and investigators in the radiation and cancer therapy field in general. Several issues warrant further clarification from the authors.

1. Given that importance of the results from reversal learning, it would have been nice to have a separate test to confirm the results. In the absence of such confirmation, authors should discuss what future studies may be employed to strengthen the conclusion in the Discussion.

Authors: We agree, and we have added a short paragraph to the Discussion about experiments to follow up on these results.

2. It was not clear initially that two separate sets of mice were used for the voluntary wheel running/OF/Y maze and the reversal learning and that mice in the voluntary wheel running group only went through OF or Y maze, but not both arena tests. Providing these information earlier in the Methods section would have been helpful.

Authors: We regret the confusion on this point. Since multiple reviewers mentioned this point, we clarified in the methods section, results section, and in the legend for Figure 1 that mice underwent only one of the two behavioral tests.

3. Elevated zero maze was not performed (or at least the data were not provided), but was mentioned in line 89.

Authors: We apologize, this was an editing error on our part and the mention of elevated 0-maze is removed.

4. There was a mention OF and Y maze tests were performed between ZT7 and ZT10 (line 89-90). Please clarify what is ZT7 and ZT10.

Authors: We have clarified this in the methods and to the legends for Figs S1 and S3.

5. The reversal learning started on Day 5 (or 3 days after irradiation), but it was not clear how many days were devoted to the reversal phase. Was there a proficiency criterion to reach before the test was terminated?

Authors: We regret leaving this out of the methods section, so we have added that reversal learning terminated after day 8. We did not use a proficiency criterion (we generally wish to avoid arbitrary thresholds) so we recorded for 4 days, which based on published data we expected to be more than enough time to completely capture reversal learning.

6. As a related question – there were two more days of training after irradiation. Were the performance from the irradiated mice comparable to the sham controls, and how was the performance compared to the pre-irradiation phase?

Authors: We have added supplementary figures (Figs S4A and B) to display this data. We also added a note in the results section that the performance appears similar between groups and also similar to performance at the end of the pre-irradiation phase.

7. To measure performance without time constraint, the authored chose to analyze performance during the first 1600 nose poke in the reversal phase. What was the rationale for selecting 1600 nose pokes as the cutoff?

Authors: We regret the lack of clarity here in the original manuscript. For Figs 3H and 3I, we chose 1600 pokes as an arbitrary axis limit for displaying the data, but as it may be confusing, we are now displaying the entire data set on these line plots. Figure 3G shows performance over the first night. For Figure 3J, we analyze the first 1536 pokes because this was the average (mean) number of pokes across all Sham animals during the first night. 

8. It will be helpful if the authors can indicate on the timescale (Figures E and F) which segments are the dark phase and which are the light phase. Likewise, because the time required to complete 1600 nose pokes in the reversal phase is different between sham and irradiated animals, it will be helpful for the authors to provide that information.

Authors: The irradiated animals take longer to make an equivalent number of pokes during reversal learning, so we have clarified this in the text of the Results section. We have also added shaded regions to Figs 3E and F to indicate the dark phase. In doing so, we should mention that in the original submission of this manuscript the data were not properly aligned to one another due to video recordings starting at slightly different times of day. We have now realigned the data appropriately, which changes the numbers slightly, but does not affect our interpretation of the data, nor does it affect the results of any significance tests. 

9. Figures 3G and 3J are a percentage over a percentage (performance during the reversal phase over performance during the first night of training). It is also not clear if 3J uses the average performance over the entire reversal phase, or just during the first 1600 nose pokes, and normalizes it to the first 1371 nose pokes. It may be helpful in visualizing the data if the authors can plot out the performance during the first night or the first 1600 nose pokes of the reversal learning and performance of the first night or the first 1371 nose pokes, respectively, during the training phase for side-by-side comparisons. 

Authors: We regret the confusion on this point. To address this, we are no longer normalizing the data; instead we are just showing the raw performance (there is no longer a percentage over a percentage). We have also done our best to clarify that Fig 3G shows the cumulative performance over the first night, and Fig 3J show the cumulative performance over the first 1536 nose-pokes (which is the average number of nose pokes over the first night). We hope this sufficiently clarifies the results.

10. The authors put sample sizes in 2A and 3A, but did not provide the same information for Y maze, OF, and BDNF plots. For consistency, these information should be provided.

Authors: We have added sample sizes to all figures.

---

## [Decision Letter · Decision Letter 1]

15 Jun 2020

PONE-D-20-08431R1

Induction of fatigue-like behavior by pelvic irradiation of male mice alters cognitive behaviors and BDNF expression.

PLOS ONE

Dear Dr. Saligan,

Thank you for submitting your manuscript to PLOS ONE. After careful consideration, we feel that it has merit but does not fully meet PLOS ONE’s publication criteria as it currently stands. Therefore, we invite you to submit a revised version of the manuscript that addresses the points raised during the review process.

As you will note Reviewer 2 has some additional minor comments that need to be addressed before the manuscript can be accepted. When the corrections are made, they will be reviewed by me and there is no need for additional review.

We look forward to receiving your revised manuscript.

Kind regards,

Michelle M. Adams, Ph.D.

Academic Editor

PLOS ONE

Additional Editor Comments (if provided):

I have also received Reviewer 1's comments indicating that the manuscript is now suitable for acceptance.

Reviewers' comments:

Reviewer's Responses to Questions

**Comments to the Author**

1. If the authors have adequately addressed your comments raised in a previous round of review and you feel that this manuscript is now acceptable for publication, you may indicate that here to bypass the “Comments to the Author” section, enter your conflict of interest statement in the “Confidential to Editor” section, and submit your "Accept" recommendation.

Reviewer #2: (No Response)

Reviewer #3: All comments have been addressed

Reviewer #4: All comments have been addressed

2. Is the manuscript technically sound, and do the data support the conclusions?

Reviewer #2: Yes

Reviewer #3: (No Response)

Reviewer #4: Yes

3. Has the statistical analysis been performed appropriately and rigorously? 

Reviewer #2: Yes

Reviewer #3: (No Response)

Reviewer #4: Yes

4. Have the authors made all data underlying the findings in their manuscript fully available?

Reviewer #2: Yes

Reviewer #3: (No Response)

Reviewer #4: Yes

5. Is the manuscript presented in an intelligible fashion and written in standard English?

Reviewer #2: Yes

Reviewer #3: (No Response)

Reviewer #4: Yes

6. Review Comments to the Author

Reviewer #2: While the authors have been responsive, the following changes are required to the figures:

1. Fig. 2. remove panels C, E, and F to suppl data as there is no significance; put the number of mice in the figure legend; in the G panel, show only the significant correlation and mention the other two in the text or figure legend.

2. Fig. 3. remove panels E, F, H, I, J to suppl data as there is no significance; put the number of mice in the figure legend; in the D panel, show only the two significant correlation and mention the third one in the text or figure legend.

3. Fig. 4. remove panel C to suppl data; panel D, show only the significant correlations; put the mouse numbers in the figure legend.

Reviewer #3: (No Response)

Reviewer #4: The authors have satisfactorily addressed all comments. There are two minor issues that require authors' attention:

1. Figure 3C, Y axis. This is normalized data (normalized to food pellets consumed on the day before irradiation). Consequently, the Y axis should be labeled as Food Pellets (% baseline).

2. Figure S5 D&E. Here the authors label Y axis as Open Time. Is this time in the center of the OF arena? If so, Center Time should be used to be consistent with labels used in Figures 2 and S2.

3. Figure S3. Can the authors provide a scale bar for the color scheme? The descriptive language "with darker colors representing more pellets, light colors representing fewer pellets" is not very helpful.

7. PLOS authors have the option to publish the peer review history of their article (what does this mean?). If published, this will include your full peer review and any attached files.

Reviewer #2: No

Reviewer #3: No

Reviewer #4: Yes: Ting-Ting Huang

---

## [Author Response · Author response to Decision Letter 1]

17 Jun 2020

Reviewer #2: While the authors have been responsive, the following changes are required to the figures:

1. Fig. 2. remove panels C, E, and F to suppl data as there is no significance; put the number of mice in the figure legend; in the G panel, show only the significant correlation and mention the other two in the text or figure legend.

2. Fig. 3. remove panels E, F, H, I, J to suppl data as there is no significance; put the number of mice in the figure legend; in the D panel, show only the two significant correlation and mention the third one in the text or figure legend.

3. Fig. 4. remove panel C to suppl data; panel D, show only the significant correlations; put the mouse numbers in the figure legend.

Authors: We have made the requested changes, moving non-significant results to supplementary material and removing non-significant correlation values (r and p values) from the figures (though they are still written in the main text of the Results section). We have added the number of mice to the figure legend of Fig 1, where we describe the experimental design; all other figures already show the number of mice (which was requested by one of the other reviewers in the previous round of comments). The number of mice is also written in the text of both the Methods section and in the Results section. 

Reviewer #4: The authors have satisfactorily addressed all comments. There are two minor issues that require authors' attention:

1. Figure 3C, Y axis. This is normalized data (normalized to food pellets consumed on the day before irradiation). Consequently, the Y axis should be labeled as Food Pellets (% baseline).

Authors: Thank you for pointing this out. The Y-axis was correct, but the figure legend was wrong, and so we have corrected the legend. To be clear, the data are not normalized; mice often eat around 100 pellets per day. 

2. Figure S5 D&E. Here the authors label Y axis as Open Time. Is this time in the center of the OF arena? If so, Center Time should be used to be consistent with labels used in Figures 2 and S2.

Authors: You are correct, and we have changed the axis label to “Center time” and expressed it properly as a percentage.

3. Figure S3. Can the authors provide a scale bar for the color scheme? The descriptive language "with darker colors representing more pellets, light colors representing fewer pellets" is not very helpful.

Authors: We agree, and we have added a scale bar to the all the figures that are in this format. We also reworded the figure legends to be more clear about what we measured.

---

## [Editor Report · Decision Letter 2]

18 Jun 2020

Induction of fatigue-like behavior by pelvic irradiation of male mice alters cognitive behaviors and BDNF expression.

PONE-D-20-08431R2

Dear Dr. Saligan,

We’re pleased to inform you that your manuscript has been judged scientifically suitable for publication and will be formally accepted for publication once it meets all outstanding technical requirements.

Kind regards,

Michelle M. Adams, Ph.D.

Academic Editor

PLOS ONE
---

## [Editor Report · Acceptance letter]

23 Jun 2020

PONE-D-20-08431R2 

Induction of fatigue-like behavior by pelvic irradiation of male mice alters cognitive behaviors and BDNF expression. 

Dear Dr. Saligan:

I'm pleased to inform you that your manuscript has been deemed suitable for publication in PLOS ONE. Congratulations! Your manuscript is now with our production department. 

Kind regards, 

on behalf of

Dr. Michelle M. Adams 

Academic Editor

PLOS ONE